# The cost-effectiveness of the Dutch In Balance fall prevention intervention compared to exercise recommendations among community-dwelling older adults with an increased risk of falls: A randomized controlled trial

Jirini Delfgaauw[1,2☯], Maaike van Gameren[1☯], Paul B. Voorn[1,3], Daniël Bossen[3], Branko F. Olij[4], Bart Visser[1,3], Mirjam Pijnappels[1], Judith E. Bosmans[2]*

1 Department of Human Movement Sciences, Faculty of Behavioural and Movement Sciences, Amsterdam Movement Sciences Research Institute, Vrije Universiteit Amsterdam, Amsterdam, the Netherlands, 2 Department of Health Sciences, Faculty of Science, Amsterdam Public Health Research Institute, Vrije Universiteit Amsterdam, Amsterdam, the Netherlands, 3 Faculty of Health, Centre of Expertise Urban Vitality, Amsterdam University of Applied Sciences, Amsterdam, the Netherlands, 4 Consumer Safety Institute (VeiligheidNL), Amsterdam, the Netherlands

☯ These authors contributed equally to this work.
* j.e.bosmans@vu.nl

## Abstract

### Background

Falls among older adults are a growing public health issue, and are associated with injuries and increased societal costs. Therefore, implementation of effective fall prevention interventions is important. Given limited healthcare resources, evaluating the cost-effectiveness of these interventions is essential. Therefore, we aimed to evaluate the cost-effectiveness of the In Balance fall prevention intervention for community-dwelling older adults with an increased risk of falls compared to general physical activity recommendations (control) from a societal perspective.

### Methods

An economic evaluation was conducted alongside a twelve month, single-blind, multicenter randomized controlled trial. Participants were 264 non- and pre-frail community-dwelling adults aged 65 years or older with an increased fall risk. We assessed costs from a societal perspective and effects included the number of falls, fall-related injuries, and Quality-Adjusted Life Years (QALYs) based on the EuroQol Five-level questionnaire (EQ-5D-5L) and the Adult Social Care Outcomes Toolkit (ASCOT). Missing data were handled using Multiple Imputation by Chained Equations (MICE). Incremental costs and effects were estimated using Seemingly Unrelated Regressions and used to estimate Incremental cost-effectiveness ratios (ICERs).

**Data availability statement:** Only the researchers have access to the data due to privacy considerations, as the dataset contains potentially identifying participant information. An anonymised version of the data is available upon request from prof. dr. Mirjam Pijnappels (m.pijnappels@vu.nl, +31205988467). Prof. dr. Pijnappels served as the Principal Investigator, project coordinator, and project leader of the project during which this study was conducted.

**Funding:** This research was funded by a research grant (#555002018) from the Netherlands Organisation for Health Research and Development (ZonMw). The funders had no role in study design, data collection and analysis, decision to publish, or preparation of the manuscript.

**Competing interests:** The first authors (JD and MvG) are early-stage investigators employed by the Vrije Universiteit Amsterdam. MvG coordinated the data collection and JD performed the analysis of the data. JD, MvG, PV, DB, BV, MP and JB have no competing interests. One author (BO) is employed by VeiligheidNL, which is the institute that owns the In Balance intervention. VeiligheidNL helped in designing the study, but was not involved in the data analysis and did not have an influence on the results in any way. No funding has been received from a commercial organization.

**Abbreviations:** ASCOT, adult social care outcome toolkit questionnaire; BMI, body mass index; CEA, cost effectiveness analysis; CEAC, cost-effectiveness acceptability curves; CUA, cost utility analysis; ED, emergency department; EQ-5D-5L, EuroQol - 5 dimensions - 5 levels; ICER, incrementel cost-effectiveness ratio; iMCQ, iMTA medical consumption questionnaire; INMB, incremental net monetary benefit; iPCQ, iMTA productivity cost questionnaire; MICE, multiple imputations by chained equations; MMSE, mini-mental state examination; PMM, predictive mean matching; QALY, quality adjusted life year; RCT, randomized controlled trial; SUR, seemingly unrelated regression; WTP, willingness to pay.

## Results

On average, In Balance was less expensive and more effective than control, but differences were not statistically significant. ICERs indicated dominance of the intervention for prevented falls (€-14,329 per prevented fall), prevented fall-related injuries (€-14,569 per prevented injury), and QALYs based on both the EQ-5D-5L (€-168,265 per QALY gained) and ASCOT (€-135,797 per QALY gained). The probability of cost-effectiveness of In Balance compared to control was 98% at a willingness to pay (WTP) of €0 per unit of effect gained.

## Conclusions

Based on this study, we conclude that In Balance may be considered cost-effective compared to control. Future research should explore whether In Balance as part of a comprehensive fall prevention strategy is cost-effective.

## Trial registration

Research with human participants: NL9248 (registered February 13 2021, URL: https://www.onderzoekmetmensen.nl/nl/trial/26195)

## Introduction

Falls among older adults are a growing public health concern [1,2]. Annually, approximately 30% of individuals aged 65 years and above experience at least one fall every year [3]. Falls commonly lead to injuries such as bruises, hip- and upper extremity fractures, and brain- and cranial damage [4,5]. Fall-related injuries, even the less severe injuries, are associated with reduced quality of life, impaired functioning and an increased risk of subsequent falls [5,6]. Fall-related injuries of older adults led to around 100,000 emergency department visits in 2023 in the Netherlands [7]. These injuries impose substantial demands on healthcare resources and lead to high healthcare costs [1,5]. Annually, healthcare costs of fall-related injuries in the Netherlands are approximately €474.4 million [5]. A systematic review showed that national fall-related costs found in prevalence-based studies ranged between 0.85% and 1.5% of the total healthcare expenditures [8]. Considering the limited available healthcare resources and the expected increase in fall incidents due to the aging population [9], prevention of falls is of utmost importance.

Fall prevention interventions aim to prevent falls and thereby to prevent injuries, improve quality of life, and reduce costs associated with falls [5]. Previous studies have shown that such interventions are effective in reducing the number of falls and improving quality of life in community-dwelling older adults [10,11]. In the Netherlands, the 'In Balance' intervention is a widely implemented, group-based fall prevention intervention in the Netherlands that includes exercises based on Tai Chi and offers education on risk factors associated with falls [12]. Research by Faber and colleagues in 2006 [13] demonstrated the effectiveness of the original 20-week In

Balance intervention for pre-frail older adults living in nursing homes. Since then, an adapted version of the In Balance intervention with a duration of 14 weeks is implemented throughout the Netherlands that targets both non-frail and pre-frail community-dwelling older adults. We recently showed that there were no statistically significant differences in falls and fall-related injuries between this adapted In Balance program and usual care, although there were less falls and fall-related injuries in the In Balance group [14].

Besides effectiveness, it is important to establish the cost-effectiveness of fall prevention interventions, for example for policy makers to optimally allocate the scarce resources available and to decide about the reimbursement of such interventions. A recent systematic review indicates that the cost-effectiveness of multifactorial fall prevention programs is uncertain with substantial variation observed in incremental cost-effectiveness ratios (ICERs) [15]. Additionally, the (cost-)effectiveness of fall prevention interventions differs between subgroups based on characteristics, such as age, health status, frailty, and the frequency of falls [13,15–17]. A possibility to make fall prevention more efficient is by providing group-based fall prevention interventions that address multiple factors associated with falling rather than individual and/or mono-factorial programs. In Balance is such a program.

The cost-effectiveness of In Balance has not been researched before, while this information is essential to assess whether the program provides value for money. Demonstrating cost-effectiveness could support further implementation and strengthen the case for broader reimbursement by health insurers, and may also support decision-making when choosing between different fall prevention programs. Therefore, our aim was to evaluate the cost-effectiveness of the nationally implemented In Balance intervention for non-frail and pre-frail community-dwelling older adults in comparison with general physical activity recommendations (control) from a societal perspective [18]. Physical activity recommendations were used as a usual care to maintain participant engagement without expecting behavioral change. We hypothesized that the In Balance intervention will lead to lower costs and increased quality of life compared to physical activity recommendations among both non- and pre-frail community-dwelling older adults with an increased risk of falls. Additionally, we expected In Balance to be more cost-effective compared to control in the pre-frail group than in the non-frail group.

## Methods

### Study design

This economic evaluation was conducted alongside a 12 month, single-blind, multicenter randomized controlled trial (RCT), which compared the cost-effectiveness of the In Balance fall prevention intervention (intervention) to written general recommendations on physical activity (control). The study was conducted according to the guidelines of the Declaration of Helsinki, and approved by the Medical Ethical Committee Brabant (project number P2055) on 10 February 2021. All participants signed written informed consent and were aware that participation is voluntary. For an in-depth description of the study design, we refer to the study protocol and the preprint of the effectiveness study [12,14]. The participant recruitment period started on the 30th of August 2021 and finished on the 18th of January 2023. The data collection ended on the 18th of January 2024. All participants provided written informed consent.

### Participants

Community-dwelling non-frail and pre-frail adults aged 65 years and above who had an increased risk of falls, were eligible to participate. A person was considered at increased risk of falling if they had experienced a minimum of two falls in the past 12 months and/or had mobility impairments (i.e., difficulties with moving, balance or walking) [19]. Frailty status was determined based on the indicators according to Fried et al. [20], which comprise unintentional weight loss, weak grip strength, slow gait speed, low physical activity, and exhaustion. Participants who met none of the criteria were classified as non-frail, whereas those meeting one or two criteria were classified as pre-frail. Participants who met three criteria or more were classified as frail and excluded from the study, since previous research found an increased fall risk in frail older

adults after participating in In Balance [13]. Other exclusion criteria were the inability to walk 100 meters independently or to perform activities of daily living without assistance, cognitive impairment defined as a score lower of 18 or lower on the Mini-Mental State Examination [21], self-reported contraindications (i.e., cardiovascular, neurological and orthopedic problems), participation in a fall prevention intervention within the last 6 months, or insufficient proficiency in Dutch, as the In Balance intervention and all materials were only available in Dutch.

## Recruitment, randomization, stratification, blinding and treatment allocation

Recruitment was done via flyers, advertisements in local papers and the network of the In Balance therapists. Participants were randomized at a 1:1 ratio to the intervention or control group, with stratification based on frailty status. The allocation sequence was generated online using Sealed Envelope by MvG. Sealed randomization envelopes in blocks of 10 (5 envelopes per group) were prepared. Participants were enrolled and assigned to a group by one of the researchers by opening an envelope. All investigators and assessors involved in this study were blinded to group assignment during the data collection. Due to the nature of this study, participants and therapists could not be blinded to group allocation.

## Intervention and control group

In Balance is a group-based fall prevention intervention with a duration of 14 weeks. It is provided by certified physical- and exercise therapists. Exercises based on Tai Chi are combined with education to improve balance and strength as well as to increase awareness of postural imbalance and fall risk (16). The intervention consists of three phases: an information phase (one week), an educational phase (three weeks), and an exercise phase with two training sessions per week (10 weeks). Participants in the intervention group received only the In Balance program as part of the study protocol and were not additionally provided with other materials on physical activity. For a detailed description of the intervention, we refer to van Gameren et al. [12].

The control group received a flyer containing advice on physical activity levels, strength and balance for older adults. For example, a minimal amount of 150 minutes of moderate to vigorous physical activity per week was recommended, distributed over several days. Also, muscle and bone strengthening activities such as walking stairs and balance exercises at least twice per week are advised. Moreover, the health benefits associated with physical activity are explained. These recommendations are according to the physical activity guidelines of the World Health Organization [22].

## Outcome measures

**Effects.** The primary outcome measures in this trial were the number of falls and the number of injuries due to a fall. A fall was defined as an unintentional descent to the ground or a lower level [13]. We assessed the occurrence of falls and fall-related injuries during the 12-month follow-up period with fall diaries and monthly follow-up telephone calls. When a fall event was reported, detailed information regarding its consequences was systematically collected from the participants to obtain information on the fall-related injuries.

The outcome measures in the economic evaluation were Quality-Adjusted Life Years (QALYs). This metric combines quality and quantity of life [23]. Quality of life was measured using the EuroQol Five-level questionnaire (EQ-5D-5L) and the Adult Social Care Outcomes Toolkit (ASCOT). The EQ-5D-5L and ASCOT were both assessed at baseline, and at months 4, 8 and 12 of follow-up. We incorporated the ASCOT additionally to the EQ-5D-5L, due to its broader scope in assessing quality of life [24]. The resulting EQ-5D-5L and ASCOT health states were converted into utility scores using the Dutch EQ-5D-5L and ASCOT tariffs [25,26]. Utility scores are anchored at 0 (death) and 1 (full health) and are multiplied with the time spent in a specific health state. Changes in health states between timepoints were linearly interpolated. The maximum number of QALYs to be experienced per person was 1 due to the total of 1 year follow-up in the study.

**Costs.** Costs were assessed using the iMTA Medical Consumption Questionnaire (iMCQ) and the iMTA Productivity Cost Questionnaire (iPCQ) at 4, 8, and 12 months after the baseline assessment. The iMCQ was used to measure healthcare costs (including primary and secondary care, home care, and medication), and patient and family costs (informal care). Costs in other sectors were measured using the iPCQ and included costs due to lost productivity (i.e., absenteeism from paid and unpaid work, and presenteeism from paid work). Although most participants were retired, 14 were still employed and 22 reported being unable to perform unpaid work due to health issues. Given the Dutch recommendation to take a broad societal perspective, productivity losses, both paid and unpaid, were included to provide a comprehensive estimate of costs [27]. Healthcare utilization was valued using standard costs from the Dutch guideline for economic evaluations [27]. Medication costs were determined based on the average daily price for generic medications from the Dutch Healthcare Institute [28]. Participants were instructed to report only their prescribed extramural medication. For valuing absenteeism from unpaid work, for example (voluntary) work or informal care, we used a shadow price (i.e., the wage rate for a legally employed cleaner) from the Dutch guideline for economic evaluations [27]. To calculate absenteeism from paid work, we applied the friction cost method [29]. This method assumes that absent workers are replaced after the friction period and that productivity is restored after that [30]. The duration of the friction period used in this study is 115 days [27]. We valued absenteeism from paid work and presenteeism using the Dutch average gross hourly wages from 2022 [27]. Participants reported their perceived efficiency while working with health complaints, referred to as the efficiency score. Lost productivity was calculated using the formula: $(1 - \text{efficiency score}) \times$ number of days with health complaints $\times$ hours worked per day. This value was then monetized based on gender-specific wage rates. All standard prices were adjusted for inflation to reflect the prices from 2022 using the consumer price index (CPI) from Statistics Netherlands [31]. Discounting costs was unnecessary due to the trial's 12-month follow-up period. The costs of the In Balance intervention were estimated according to a template of VeiligheidNL, and divided among the maximum number of 12 participants per training group, see S1 Table for an overview of this calculation.

### Statistical analysis

The analysis consisted of a cost-effectiveness analysis (CEA) and a cost-utility analysis (CUA) according to the intention-to-treat (ITT) principle. The analyses were carried out using RStudio (Version 2023.12.1 + 402).

### Sample size

Based on the expected effect size of a 50% decrease in the number of falls in the intervention group compared to the control group, a minimum of 106 individuals per group was needed. The expected 50% reduction in falls was chosen to reflect a clinically relevant effect and was also based on a study of Faber and colleagues, who used a similar estimate [13]. This calculation assumed a statistical power of 0.80, a type II error rate (β) of 0.20, and a significance threshold (α) of 0.05. Taking into account an expected dropout rate of 20%, we aimed to include 256 participants. Due to the need for a sufficient number of participants to start a new In Balance training group in this study, we included a total of 264 participants.

### Data imputation

To impute missing observations for costs and effects, we used Multiple Imputations by Chained Equations (MICE) with Predictive Mean Matching (PMM) [32,33]. PMM replaces missing values by sampling only from the distribution of the observed data and adds a random element to reflect uncertainty [32,33]. PMM is especially suitable for handling non-normally distributed data such as costs that are generally right skewed [32,33]. The number of imputations was increased until the loss of efficiency was less than 5%, resulting in 20 imputed datasets [32]. The cost-effectiveness analyses as described below were performed on each imputed dataset, after which results were pooled using Rubin's rules [34].

The rate of missing data was 10.4% for fall diaries and follow-up calls, and 16.5% for cost and quality of life questionnaires in the intervention group. In the control group, the percentages of missing data were 15.0% and 26.1%, respectively.

Participants with complete data had statistically significantly higher Mini-Mental State Examination scores (indicating better cognitive functioning) and a lower fall history at baseline. Given these differences between participants with complete and incomplete data, we conducted MICE under the Missing at Random (MAR) assumption. To strengthen the plausibility of this assumption, both MMSE scores and fall history were included as predictors in the imputation model [32].

### Descriptive variables

All descriptive variables were collected using questionnaires. Physical activity, however, was assessed with an inertial sensor (DynaPort MoveMonitor Plus, McRoberts BV, The Netherlands) [4,5]. The sensor was worn on the lower back for seven consecutive days, preferably day and night except during water activities, and was returned by mail afterwards.

### Main analysis

Cost and effect differences between the intervention and control group were estimated using Seemingly Unrelated Regression (SUR) analysis which preserves the correlation between costs and effects. To apply SUR, all outcomes (including the number of falls) were treated as continuous variables. Age and gender were included as confounders in all analyses [35], and physical activity was included in the analyses with falls and fall-related injuries as effect outcome [10]. Other baseline characteristics were considered confounders if the estimate for the randomization group changed by at least 10% after adding that variable to the model. The confounders can be found in the footnote below the tables in the results section. We calculated ICERs by dividing the differences in costs by the differences in effects. To minimize reliance on distributional assumptions of continuous outcome scales, we estimated the uncertainty surrounding the ICERs using bias-corrected accelerated bootstrapping with 5,000 replications [36]. The 95% Confidence intervals were derived from the variance of the bootstrapped regression coefficients. ICERs and their associated uncertainty were presented on the Cost-Effectiveness Plane (CE-plane). Moreover, the Incremental Net Monetary Benefit (INMB) approach was applied to estimate Cost-Effectiveness Acceptability Curves (CEAC). For falls and fall-related injuries, we used a willingness-to-pay (WTP) threshold of €10,000 per fall-related injury prevented, based on the average healthcare costs of €9,370 per fall after an ED visit [1]. For QALYs, a WTP threshold was used of €50,000 per QALY gained, based on the currently applied range in the Netherlands [37].

### Secondary analyses

In addition to the main analysis, we performed an a priori defined stratified analysis for frailty status (i.e., non- and pre-frail). Furthermore, we conducted three sensitivity analyses to assess the robustness of the findings. For the first sensitivity analysis (SA1), we conducted a per-protocol analysis. This analysis included only participants who complied with the designated protocols, excluding those with less than 75% attendance in the intervention group and those in the control group who participated in any fall prevention intervention [38]. In the Netherlands the societal perspective is the preferred perspective, meaning that all costs are considered regardless of who bears them [39]. The advantage of using the societal perspective over narrower perspectives such as the healthcare perspective is that (unexpected) shifts in costs between sectors can be identified [18]. Some other countries make decisions from a healthcare perspective. To inform health insurers and to be able to compare results of this study with literature, the second sensitivity analysis (SA2) was conducted from a healthcare perspective, meaning that we excluded lost productivity costs, and patient and family costs [40]. For the third sensitivity analysis (SA3), we conducted a complete case analysis to assess the robustness of the findings to the missing-at-random assumption and to explore the potential risk of bias due to missing data [33].

### Adverse events

Adverse events were defined as any undesirable experience occurring to a participant during the study, whether or not considered related to the In Balance intervention or the trial procedure. All adverse events reported spontaneously by the

participant or observed by the investigator or staff were recorded. Occurrence of adverse events was assessed during the monthly telephone calls and for the intervention group during the In Balance intervention by the therapist.

## Results

### Participants

Between June 2021 and January 2023, 849 potential participants were screened for eligibility (Fig 1). Of these, 264 people were included in the study between August 2021 and January 2023 and were randomly assigned to the intervention group (n = 131) or the control group (n = 133). Follow-up finished in January 2024.

In the intervention group, 40 (30%) participants were classified as non-frail and 91 (70%) as pre-frail. In the control group, 37 (28%) participants were non-frail and 96 (72%) were pre-frail (Fig 1). No clinically relevant differences in baseline characteristics were observed between the intervention and control groups (Table 1). Likewise, there were no relevant baseline differences between groups after stratifying for frailty or in the per-protocol sample (S2 and S3 Tables).

### Costs

Table 2 presents the mean costs in the two groups and the unadjusted differences in mean costs between groups. The costs of the In Balance intervention were estimated at €252 per person. From the societal perspective, mean costs in the

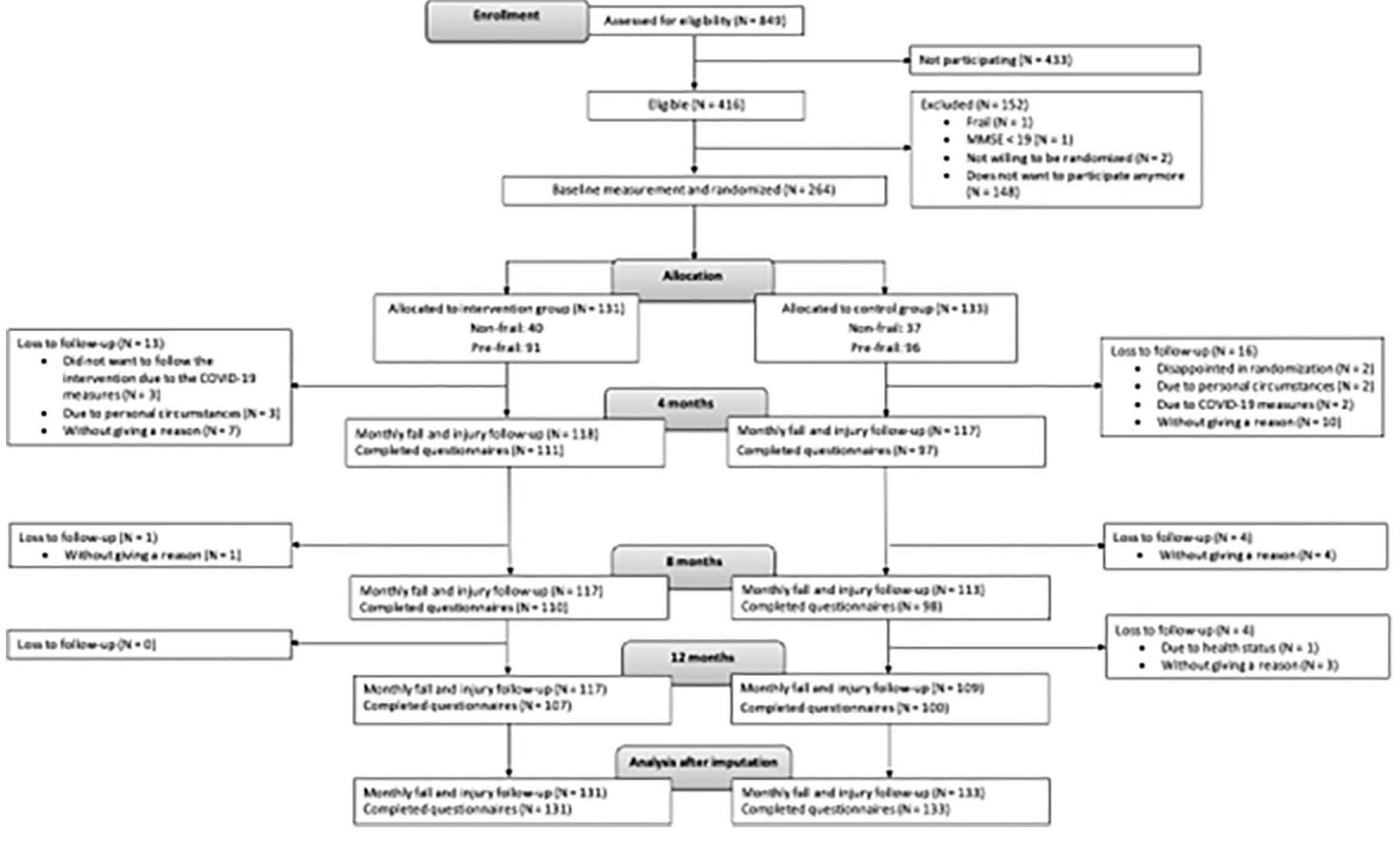

**Fig 1. Flowchart of the numbers of inclusion and loss to follow-up.**

---

**Table 1. Baseline characteristics per treatment group. Characteristics are presented as n (%) unless specified otherwise.**

| Variable | In Balance group (N = 131) | Control group (N = 133) |
|---|---|---|
| Age (years, SD) | 75.19 (5.60) | 75.70 (5.12) |
| Gender (female) | 99 (76%) | 102 (77%) |
| Body Mass Index (kg/m², SD) | 26.52 (4.22) | 26.79 (5.01) |
| Mini-Mental State Examination (score, SD) | 27.65 (2.19) | 27.48 (2.29) |
| Marital status | | |
| Lawfully married/living together | 58 (45%) | 61 (50%) |
| Unmarried/divorced/widowed | 70 (55%) | 61 (50%) |
| Having children | 96 (75%) | 92 (74%) |
| Living alone | 72 (56%) | 66 (53%) |
| Education | | |
| Low | 0 (0%) | 7 (6%) |
| Moderate | 38 (30%) | 31 (25%) |
| High | 90 (70%) | 86 (69%) |
| Smoking | 6 (5%) | 7 (6%) |
| Use of alcohol | 91 (71%) | 92 (74%) |
| Use of different medications per week (SD) | 3.07 (3.45) | 3.77 (6.40) |
| Dizziness | 36 (28%) | 27 (23%) |
| Incontinence | 58 (45%) | 65 (52%) |
| Number of falls in previous year | | |
| None/once | 70 (56%) | 72 (58%) |
| Twice or more | 56 (44%) | 52 (42%) |
| Use of aids | | |
| Walking | 23 (18%) | 19 (15%) |
| Vision | 126 (98%) | 118 (95%) |
| Hearing | 27 (21%) | 30 (24%) |
| Attending physiotherapy | 44 (35%) | 40 (32%) |
| Physical activity per day (SD) | | |
| Number of hours being physically active | 1.39 (0.64) | 1.41 (0.59) |
| Number of steps | 6619 (3387) | 6559 (3102) |
| EQ-5D-5L Baseline utility (score, SD) | 0.81 (0.13) | 0.79 (0.16) |
| ASCOT Baseline utility (score, SD) | 0.86 (0.14) | 0.85 (0.14) |

Note. SD = Standard deviation, EQ-5D-5L = EuroQol questionnaire, ASCOT = Adult Social Care Outcome Toolkit questionnaire.

intervention group were €6,170, compared to €10,265 in the control group. For all cost categories, with exception of the intervention costs and medication costs, mean costs in the intervention group were lower than in the control group. However, none of the cost differences reached statistical significance.

## Effect outcomes

Table 2 presents the number of falls, fall-related injuries, and QALYs in the two groups and the unadjusted differences between groups. For all effect outcomes, the differences were in favor of the intervention group. However, none of the effect differences were statistically significant, except for the difference in QALYs based on the ASCOT.

**Table 2. The pooled mean effects, mean costs and differences per treatment group.**

| Outcome | In Balance (SE) | Control (SE) | Difference (95%CI) |
|---|---|---|---|
| | n = 131 | n = 133 | |
| **Costs in €** | | | |
| **Healthcare costs** | 5,002 (772) | 7,771 (2197) | −2,769 (−7221; 1,681) |
| **Primary care costs** | 833 (94) | 1,037 (169) | −204 (−566; 156) |
| **Secondary care costs** | 2,144 (573) | 4,601 (2160) | −2,457 (−6,658; 1744) |
| **Medication costs** | 1,097 (375) | 1,067 (270) | 30 (−872; 931) |
| **Home care costs** | 675 (121) | 1,064 (299) | −389 (−987; 208) |
| **Intervention costs** | 252 | 0 | 252 |
| **Informal care costs** | 430 (127) | 623 (204) | −193 (−655; 269) |
| **Lost productivity costs** | 737 (142) | 1,869 (469) | −1132 (−2,706; 442) |
| **Total societal costs** | 6,170 (830) | 10,265 (2526) | −4,095 (−9,239; 1,049) |
| **Effects** | | | |
| **Average number of falls** | 1.67 (0.24) | 1.98 (0.37) | −0.30 (−1.22; 0.62) |
| **Average number of fall- related injuries** | 0.70 (0.11) | 0.97 (0.18) | −0.27 (−0.68; 0.14) |
| **QALY (EQ-5D-5L)** | 0.82 (0.01) | 0.78 (0.03) | 0.04 (−0.02; 0.10) |
| **QALY (ASCOT)\*** | 0.88 (0.01) | 0.84 (0.02) | 0.04 (0.003; 0.078) |

Notes. SE = Standard error, 95%CI = 95% Confidence interval, QALY = Quality adjusted life year, EQ-5D-5L = EuroQol questionnaire, ASCOT = Adult Social Care Outcome Toolkit questionnaire. Costs and effect differences are crude estimates. The intervention costs are estimated per person. Clinical effects are presented in the number of falls and injuries (not in prevented falls and prevented fall-related injuries). *Significantly different between groups.

## Cost-effectiveness analysis

Table 3 presents cost-effectiveness results for falls, fall-related injuries, and QALYs from a societal perspective. The adjusted cost difference in total societal costs was €- 3,991 (95%CI: −8,993–1,011) between the intervention and the control groups. On average, the intervention was more effective and less expensive, i.e., dominant, over control for all four effect outcomes, although differences were not statistically significant.

The In Balance intervention prevented on average 0.28 falls compared to the control (95% CI: −0.64 to 1.20). The ICER was €-14,329 per prevented fall with 70% of the bootstrapped cost-effect pairs in the southeast quadrant of the cost-effectiveness plane (more effective and less expensive) (Fig 2A). The probability of cost-effectiveness of the intervention compared to the control group was 98% at a WTP threshold of €0 per prevented fall and 91% at a WTP threshold of €10,000 per prevented fall (Fig 2B).

On average, the number of prevented fall-related injuries was 0.27 (95% CI: −0.14; 0.69) for the intervention compared to the control group. The ICER was €-14,569 per prevented injury, with 89% of the bootstrapped cost-effect pairs in the southeast quadrant of the cost-effectiveness plane (S4 Fig 1A). The probability of cost-effectiveness of the intervention compared to the control group was 98% at a WTP threshold of €0 per prevented injury and 98% at a WTP threshold of €10,000 per prevented injury (S4 Fig 1B).

For QALYs based on the EQ-5D-5L, the intervention group gained 0.02 QALYs (95% CI: −0.03 to 0.08) compared to the control group. The ICER was €-168,265 per QALY gained, with 79% of the bootstrapped cost-effect pairs in the southeast quadrant of the cost-effectiveness plane (more QALYs, less costly) (Fig 2C). The probability that the intervention was cost-effective compared to the control was 98% at a WTP threshold of €0 per QALY gained and 98% at a WTP threshold of €50,000 per QALY gained (Fig 2D).

**Table 3. Cost and effect differences, ICERs and distributions on the cost-effectiveness plane for the main and sensitivity analyses.**

| Outcomes | In Balance (N) | Control (N) | ΔC (95%CI) | ΔE (95%CI) | ICER | Distribution CE-plane | | | |
|---|---|---|---|---|---|---|---|---|---|
| | | | € | Unit of effect | €/unit of effect | NE | SE | SW | NW |
| **Main analysis** | | | | | | | | | |
| **Prevented falls[a]** | 131 | 133 | −3,991 (−8,993; 1,011) | 0.28 (−0.64; 1.20) | − 14,329 Dominant | 1% | 70% | 28% | 1% |
| **Prevented fall-related injuries[b]** | 131 | 133 | −3,991 (−8,987; 1,006) | 0.27 (−0.14; 0.69) | −14,569 Dominant | 2% | 89% | 9% | 0% |
| **QALY (EQ-5D-5L)[c]** | 131 | 133 | −3,991 (−9,012; 1,030) | 0.02 (−0.03; 0.08) | −168,265 Dominant | 2% | 79% | 19% | 0% |
| **QALY (ASCOT)[d]** | 131 | 133 | −3,991 (−8,999; 1,017) | 0.03 (−0.002; 0.061) | −135,797 Dominant | 2% | 96% | 2% | 0% |
| **SA1: Per Protocol analysis** | | | | | | | | | |
| **Prevented falls[a]** | 54 | 130 | −4,492 (−10,185; 1,200) | 0.05 (−1.0; 1.1) | −91,978 Dominant | 1% | 52% | 45% | 2% |
| **Prevented fall-related injuries[b]** | 54 | 130 | −4,492 (−10,173; 1,189) | 0.24 (−0.26; 0.74) | −18,603 Dominant | 2% | 81% | 16% | 1% |
| **QALY (EQ-5D-5L)[c]** | 54 | 130 | −4,492 (−10,176; 1,192) | 0.04 (−0.01; 0.1) | −101.784 Dominant | 3% | 93% | 4% | 0% |
| **QALY (ASCOT)*[d]** | 54 | 130 | −4,492 (−10,184; 1,199) | 0.04 (0.01; 0.08) | −104,408 Dominant | 3% | 97% | 0% | 0% |
| **SA2: Healthcare perspective** | | | | | | | | | |
| **Prevented falls[a]** | 131 | 133 | −2,700 (−7,020; 1,619) | 0.28 (−0.64; 1.19) | −9,750 Dominant | 4% | 67% | 26% | 3% |
| **Prevented fall-related injuries[b]** | 131 | 133 | −2,700 (−7,031; 1,630) | 0.27 (−0.14; 0.69) | −9,856 Dominant | 6% | 84% | 9% | 1% |
| **QALY (EQ-5D-5L)[c]** | 131 | 133 | −2,700 (−7,019; 1,618) | 0.02 (−0.03; 0.08) | −115,673 Dominant | 5% | 75% | 18% | 2% |
| **QALY (ASCOT)[d]** | 131 | 133 | −2,700 (−7,020; 1,619) | 0.03 (−0.002; 0.061) | −92,300 Dominant | 7% | 91% | 2% | 0% |
| **SA3: Complete case analysis** | | | | | | | | | |
| **Prevented falls[a]** | 52 | 38 | −832 (−3,476; 1,812) | 0.31 (−0.60; 1.23) | −2,656 Dominant | 20% | 58% | 14% | 8% |
| **Prevented fall-related injuries[b]** | 52 | 38 | −832 (−3481; 1,817) | 0.22 (−0.14; 0.57) | −3,811 Dominant | 24% | 65% | 8% | 3% |
| **QALY (EQ-5D-5L)[c]** | 52 | 48 | 7 (−2344; 2,358) | 0.008 (−0.016; 0.03) | 877 Trade-off | 35% | 37% | 12% | 16% |
| **QALY (ASCOT)[d]** | 50 | 48 | 91(−2235; 2,416) | 0.02 (−0.005; 0.041) | 5,125 Trade-off | 49% | 45% | 2% | 4% |

Notes. C = costs, E = effects, ICER = Incremental Cost Effectiveness Ratio, CE-plane = Cost-effectiveness plane, 95%CI = 95% Confidence interval, SA = sensitivity analysis, NE = Northeast, SE = Southeast, SW = Southwest, NW = Northwest, Dominant = more effective and less costly. Main analysis according to the intention-to-treat principle. [a] Adjusted for age, sex, physical activity and emotional wellbeing. [b] Adjusted for age, sex and physical activity. [c] Adjusted for age, sex, baseline utility and emotional wellbeing. [d] Adjusted for age and sex, baseline utility and emotional wellbeing. *Significantly different between groups.

For QALYs based the ASCOT, the difference in QALYs was 0.03 in favor of In Balance (95% CI: −0.002 to 0.061). The ICER was €-135,797 per QALY gained, with 96% of the bootstrapped cost-effect pairs in the southeast quadrant of the cost-effectiveness plane (S5 Fig 1A). The probability of cost-effectiveness of the intervention compared to the control was 98% at WTP threshold of €0 per QALY gained and 99% at a WTP threshold of €50,000 per QALY gained (S5 Fig 1B).

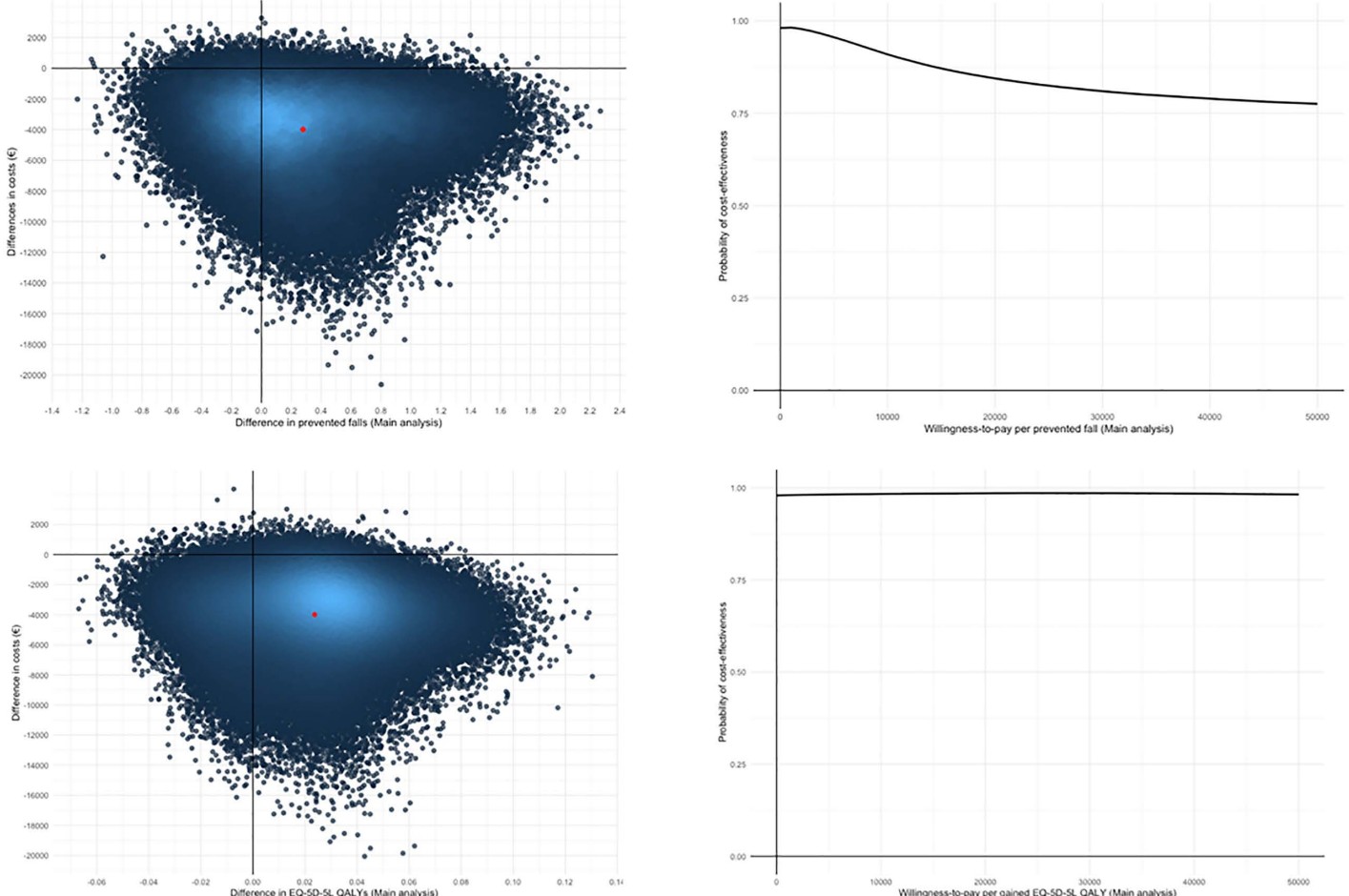

**Fig 2. Cost-effectiveness planes and cost-effectiveness acceptability curves.** A. Cost-effectiveness plane of the main analysis for prevented falls. B. Cost-effectiveness acceptability curve of the main analysis for prevented fall. C. Cost-effectiveness plane of the main analysis for QALYs based on the EQ-5D-5L. D. Cost-effectiveness acceptability curve of the main analysis for QALYs based on the EQ-5D-5L.

### Sensitivity analyses

Table 3 also presents the results for SA1 (per-protocol analysis), SA2 (healthcare perspective) and SA3 (complete case analysis. For SA1, 54 In Balance participants and 130 participants of the control group adhered to the protocol criteria and were included in the analysis. In the per protocol analysis, the cost difference was slightly larger compared to the main analysis (€-4,492 in the per protocol-analysis vs €-3,991 in the main analysis). Although the number of prevented falls by In Balance compared to the control became smaller in the per-protocol analysis (0.05) compared to the main analysis (0.28), the number of prevented fall-related injuries in the two analyses was similar. The difference in QALYs was similar in the per protocol and main analyses, although the difference in QALYs based on the ASCOT became statistically significant in the per protocol analysis. The probability of cost-effectiveness at a WTP of €0 per unit of effect gained (97%) was similar to the main analysis (98%).

The cost difference from the healthcare perspective (€-2,700 (95% CI: – 7,020; 1,619) was smaller than the difference from the societal perspective (€-3,991) while the effect differences were the same. As a result, the probability of

cost-effectiveness from the healthcare perspective compared to the societal perspective was lower (93% vs 98%, respectively) at a WTP threshold of €0 per unit of effect gained.

In SA3, between 90 and 100 participants with complete data were included, depending on the outcome measure. For the effect outcomes, the complete case analysis showed similar results compared to the main analysis. However, the difference in societal costs between the intervention and control group changed, with the difference being closer to zero and the confidence intervals narrower (table 3). This resulted in a lower probability of cost-effectiveness across all outcomes.

### Subgroup analysis

Table 4 presents the results for non-frail and pre-frail participants separately. Overall, results were similar to the main analysis with lower costs and better effects in the intervention group compared to control, although differences were not statistically significant. However, the cost difference in the pre-frail group (€-3,934) was slightly larger than in the non-frail group (€-3,774), whereas effect differences were slightly smaller in the pre-frail group than in the non-frail group. Overall, the conclusions regarding the cost-effectiveness of the intervention compared to control did not change.

### Adverse events

No severe adverse events happened that were related to the intervention during this study.

### Discussion

The aim of this study was to evaluate the cost-effectiveness of the In Balance fall prevention intervention in comparison with written general physical activity recommendations among community-dwelling non-frail and pre-frail older adults with

**Table 4. Cost and effect differences, ICERs and distributions on the cost-effectiveness plane for the subgroup analysis stratified for frailty status.**

| Outcome | ΔC (95%CI) | ΔE (95%CI) | ICER | Distribution CE-plane | | | |
|---|---|---|---|---|---|---|---|
| | € | Unit of effect | €/unit of effect | NE | SE | SW | NW |
| **Subgroup analysis: Pre-frail – In Balance (N=91), Control (N=96)** | | | | | | | |
| **Prevented falls[a]** | −3,934 (−9,879; 2,009) | 0.25 (−0.89; 1.39) | −15,559 Dominant | 3% | 61% | 33% | 3% |
| **Prevented fall-related injuries[b]** | −3,934 (−9,852; 1,982) | 0.15 (−0.35; 0.65) | −26,306 Dominant | 4% | 69% | 25% | 2% |
| **QALY (EQ-5D-5L)[c]** | −3,934 (−9,844; 1,974) | 0.02 (−0.05; 0.08) | −223,746 Dominant | 4% | 69% | 25% | 2% |
| **QALY (ASCOT)[d]** | −3,934 (−9,864; 1,994) | 0.03 (−0.01; 0.07) | −122,036 Dominant | 6% | 90% | 4% | 0% |
| **Subgroup analysis: Non-frail – In Balance (N=40), Control (N=37)** | | | | | | | |
| **Prevented falls[a]** | −3,774 (−10,072; 2,523) | 0.36 (−0.50; 1.23) | −10,360 Dominant | 2% | 78% | 19% | 1% |
| **Prevented fall-related injuries*[b]** | −3,774 (−10,052; 2,504) | 0.57 (0.07; 1.10) | −6,667 Dominant | 3% | 96% | 1% | 0% |
| **QALY (EQ-5D-5L)[c]** | −3,774 (−10,077; 2,529) | 0.04 (−0.01; 0.09) | − 88,150 Dominant | 3% | 95% | 2% | 0% |
| **QALY (ASCOT)[d]** | −3,774 (−10,072; 2,524) | 0.02 (−0.01; 0.06) | −158,802 Dominant | 3% | 89% | 8% | 0% |

Notes. C = costs, E = effects, ICER = Incremental Cost Effectiveness Ratio, CE-plane = Cost-effectiveness plane, 95%CI = 95% Confidence interval, SA = sensitivity analysis, NE = Northeast, SE = Southeast, SW = Southwest, NW = Northwest, Dominant = more effective and less costly. [a] Adjusted for age, sex, physical activity and emotional wellbeing. [b] Adjusted for age, sex and physical activity. [c] Adjusted for age, sex, baseline utility and emotional wellbeing. [d] Adjusted for age and sex, baseline utility and emotional wellbeing. *Significantly different between groups.

an increased fall risk. Our findings indicate that the In Balance intervention was dominant (more effective and less costly) over general physical activity recommendations, although the differences in total societal costs and effects were not statistically significant. The probability that the In Balance intervention is cost-effective compared to the control is 98% at a WTP threshold of €0 per unit of effect gained in the main analysis. Results stratified for non- and pre-frail older adults did not differ from the results for the total group.

Our study aligns with other recent trial-based CEAs and CUAs of multifactorial fall prevention interventions, which in general show no statistically significant differences in cost and effect outcomes [16,17,41–43]. The differences in effect outcomes in our study were slightly more positive than in these previous studies. Also, our study showed that costs in the In Balance group were lower than in the control group, which contrasts with previous studies [16,17,41]. Other Dutch cost-effectiveness evaluations by Hendriks and colleagues (2008) and Peeters and colleagues (2011) of (multidisciplinary) fall prevention interventions also found no statistically significant differences in falls, costs, and quality of life, similar to our findings [42,43]. However, both studies concluded that their intervention were not cost-effective, whereas In Balance showed a 98% probability of being cost-effective due to lower costs in the In Balance group [42,43]. This difference may relate to the inclusion of participants and nature of the intervention: Hendriks et al. evaluated medical and occupational assessments (e.g., assessments of general physical examination and daily functioning) [42], and Peeters et al. focused on multifactorial evaluation and treatment of persons with a high risk of recurrent falls [43], while In Balance combines exercise and education, which is an approach more generally proven to reduce falls [44].

The more favorable results in our study compared to international studies may be attributed to the intervention's greater intensity—training participants for two hours per week—compared to the monthly in-home visits by fall-prevention specialists in the study by Isaranuwatchai et al [16]. Compared to Matchar et al, our study population appears to be healthier and less frail, a factor they suggest is associated with a higher probability of cost-effectiveness [17]. However, this is not confirmed by our study, since the number of prevented falls and fall-related injuries, and societal costs were similar in the non-frail and pre-frail groups.

Since specific WTP thresholds for prevented falls and injuries have not yet been established by policy makers and health institutions, we used a WTP threshold of €10,000 per prevented fall-related injury, based on the estimated average healthcare costs of €9,370 per fall among individuals who visit the emergency department because of a fall-related injury [1]. However, this WTP threshold might be too high, since fall-related injuries predominantly consist of minor injuries [4,45], and costs associated with fall-related injuries are generally lower than those requiring treatment at an emergency department [8].

According to the Dutch Integral Care Agreement (ICA), municipalities are responsible for implementing fall prevention interventions within a chain approach [46]. This approach includes detection, screening, intervention application, and follow-up exercise interventions to prevent falls. Although the small and statistically non-significant effects on falls and injuries of In Balance in this study should be borne in mind, the high probability of cost-effectiveness may indicate that In Balance is a promising intervention to include in the chain approach [47]. Future research should explore how the effectiveness of In Balance can be increased, but also whether In Balance as part of a comprehensive fall prevention strategy is cost-effective, considering its integration within the entire chain approach.

A strength of this study was that this economic evaluation was conducted alongside a randomized controlled trial across the Netherlands, in large cities and in small villages, providing a good representation of the context and population of the In Balance intervention. This highlights the need for additional efforts to also include older adults with for example lower educational levels and migration backgrounds in the In Balance intervention to ensure broader societal representation. Second, this study included several outcome measures such as falls, fall-related injuries and quality of life, resulting in a comprehensive view of the intervention's effectiveness. Lastly, missing data were imputed by multiple imputation to take selective missing data into account to prevent bias. The complete case analysis showed notable changes in the point estimates and uncertainty of costs compared to the main analysis. These results suggest that dropout from the trial was

selective, with control group participants who had higher costs being more likely to drop out than those in the intervention group. We therefore preferred MICE, as it utilized more of the available data and likely prevented an underestimation of the cost difference between groups.

A limitation of this study was that the effect of the intervention between the two groups was smaller than the difference that we assumed in our power calculation. We expected a decrease of 50% in falls after the intervention, whereas in our study the actual decrease in falls was 17%, suggesting that the sample size might not have been large enough to detect statistically significant differences in effects [47]. Because of the skewed distribution of costs even larger sample sizes are needed, making our study underpowered to detect a significant difference in costs. Moreover, the In Balance intervention is delivered in a group setting, and intervention costs per participant were calculated based on an optimal group size of 12 participants per group. However, as several groups were smaller in practice, the actual costs per participant may have been higher.

## Conclusions

The In Balance fall prevention intervention was dominant (i.e., on average more effective and less expensive) over general physical activity recommendations (control), although the differences in costs and effects between the In Balance intervention and the control group were not statistically significant. At a willingness to pay threshold of €0 per unit of effect gained, the In Balance intervention showed a high probability of 98% cost-effectiveness, which was due to the lower mean societal costs in the In Balance group.

## Supporting information

**S1 Table. Calculation of the costs of the In Balance intervention.**
(DOCX)

**S2 Table. Baseline characteristics of the participants stratified for frailty status.**
(DOCX)

**S3 Table. Baseline characteristics of the participants per-protocol.**
(DOCX)

**S4 Fig. 1A. Cost-effectiveness plane of the main analysis for fall-related injuries.**
(TIF)

**S4 Fig. 1B. Cost-effectiveness acceptability curve of the main analysis for fall-related injuries.**
(TIF)

**S5 Fig. 1A. Cost-effectiveness plane of the main analysis for QALYs based on the ASCOT.**
(TIF)

**S5 Fig. 1B. Cost-effectiveness acceptability curve of the main analysis for QALYs based on the ASCOT.**
(TIF)

**S1 Protocol. Study protocol.**
(PDF)

**S1 Checklist. CONSORT-2010-Checklist.**
(DOC)

**S2 Checklist. CHEERS-2022-checklist-1.**
(DOCX)

## Acknowledgments

We want to thank all participants and therapists for participating in this study. We also want to thank VeiligheidNL, in particular Sanne Frazer, for their help with designing this study.

## Author contributions

**Conceptualization:** Jirini Delfgaauw, Maaike van Gameren, Paul B. Voorn, Daniël Bossen, Branko F. Olij, Bart Visser, Mirjam Pijnappels, Judith E. Bosmans.

**Data curation:** Maaike van Gameren.

**Formal analysis:** Jirini Delfgaauw.

**Funding acquisition:** Daniël Bossen, Bart Visser, Mirjam Pijnappels, Judith E. Bosmans.

**Investigation:** Jirini Delfgaauw, Maaike van Gameren, Paul B. Voorn.

**Methodology:** Maaike van Gameren, Paul B. Voorn, Daniël Bossen, Branko F. Olij, Bart Visser, Mirjam Pijnappels, Judith E. Bosmans.

**Project administration:** Mirjam Pijnappels.

**Supervision:** Daniël Bossen, Bart Visser, Mirjam Pijnappels, Judith E. Bosmans.

**Visualization:** Jirini Delfgaauw.

**Writing – original draft:** Jirini Delfgaauw.

**Writing – review & editing:** Jirini Delfgaauw, Maaike van Gameren, Paul B. Voorn, Daniël Bossen, Branko F. Olij, Bart Visser, Mirjam Pijnappels, Judith E. Bosmans.

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
