## [Decision Letter · Decision Letter 0]

3 Jun 2025

Dear Dr. Bosmans,

Thank you for submitting your manuscript to PLOS ONE. After careful consideration, we feel that it has merit but does not fully meet PLOS ONE’s publication criteria as it currently stands. Therefore, we invite you to submit a revised version of the manuscript that addresses the points raised during the review process.

We look forward to receiving your revised manuscript.

Kind regards,

Rositsa Koleva-Kolarova

Academic Editor

PLOS ONE

Journal Requirements:

“The first authors (JD and MvG) are early-stage investigators employed by the Vrije Universiteit Amsterdam. MvG coordinated the data collection and JD performed the analysis of the data. JD, MvG, PV, DB, BV, MP and JB have no competing interests. One author (BO) is employed by VeiligheidNL, which is the institute that owns the In Balance intervention. VeiligheidNL helped in designing the study, but was not involved in the data analysis and did not have an influence on the results in any way. No funding has been received from a commercial organization.”

We note that one or more of the authors are employed by a commercial company: name of commercial company.

3. In the online submission form, you indicated that [Insert text from online submission form here].

“We want to thank all participants and therapists for participating in this study. Moreover, this research was funded by a research grant (#555002018) from the Netherlands Organisation for Health Research and Development (ZonMw). The funders had no role in study design, data collection and analysis, decision to publish, or preparation of the manuscript. We also want to thank VeiligheidNL, in particular Sanne Frazer, for their help with designing this study. **“**

“This research was funded by a research grant (#555002018) from the Netherlands Organisation for Health Research and Development (ZonMw). The funders had no role in study design, data collection and analysis, decision to publish, or preparation of the manuscript.”

Additional Editor Comments (if provided):

Dear Authors,

Thank you for submitting your manuscript to Plos One.

I would like to state that this is a well-conducted economic analysis and a well-written article which I enjoyed reading.

I have now reached a decision which is Major Revision.

In addition to the comments from the reviewers, please see below my comments:

1. Could the authors please justify not including non-Dutch speakers in the study?

2. Similarly to the comment from another reviewer, could the authors please clarify why they considered productivity losses in a population group of elderly and potentially retired participants? Could the authors please elaborate if excluding productivity losses would have an impact on cost-effectiveness results?

Thank you.

Good luck with revising the article.

Reviewers' comments:

Reviewer's Responses to Questions

**Comments to the Author**

1. Is the manuscript technically sound, and do the data support the conclusions?

Reviewer #1: Yes

Reviewer #2: Yes

2. Has the statistical analysis been performed appropriately and rigorously?

Reviewer #1: Yes

Reviewer #2: Yes

3. Have the authors made all data underlying the findings in their manuscript fully available?

Reviewer #1: No

Reviewer #2: Yes

4. Is the manuscript presented in an intelligible fashion and written in standard English?

Reviewer #1: Yes

Reviewer #2: Yes

Reviewer #1: General comments:

This manuscript describes a well-conducted economic evaluation of the In Balance intervention compared with physical activity recommendations in non-frail and pre-frail older adults with a high fall risk. The manuscript reads well and the provided information is clear. Findings support the conclusion. Some clarifications would be helpful on methodological choices that were made.

Specific comments:

• Introduction (p. 4, line 65): You say that In Balance is a widely implemented intervention. Is it widely implemented in the Netherlands or Europe or the world? In line 74 you say that it is important to evaluate the cost-effectiveness to inform policy makers to optimally allocate resources or to arrange reimbursement. Given that it is already implemented, how is the intervention financed now? Are there concerns regarding implementation (e.g. are the right people reached)? In other words, why is it important to do the cost-effectiveness evaluation now that the intervention is already implemented?

• Introduction (p. 5, line 85): why was physical activity recommendations (line 85) or exercise recommendations (line 87) chosen as the control condition? Can this be considered usual care or an active (or passive?) control condition?

• Methods – Participants (p. 6, line 109): Why were frail participants excluded?

• Methods – Intervention and control group (p. 7, line 131): Could you please describe briefly what these physical activity recommendations were?

• Methods - Costs (p. 7/8): Were costs of the intervention itself measured and if so, how?

• Methods – Costs (p. 8, line 139): Given the inclusion criterion of being aged 65 years and over, and the retirement age being 67, I wonder how relevant it was to include costs of lost productivity (IPCQ). With the average age being 75 years, how many participants were in paid work? Unpaid work is probably more important in this age-range. The types of activities people do and the hours per week varies a lot. How reliable/realistic is it to use one shadow price for all these different activities?

• Methods – Sample size (p. 0): An expected effect size of 50% seems overly optimistic given effects founds in previous falls prevention trials. What was this expectation based on?

• Methods - Imputation (p. 10): What was the proportion and pattern of missing values? Later I noted that this information was provided in the Results on page 12. Perhaps consider moving this information to the methods, so that the reader understand methodological choices that were made early on.

• Results – Table 1: How were the descriptive variables measured, particularly physical activity?

• Results – Adverse events (p. 19): Please describe how adverse events were measured in the methods.

• Discussion (p. 20, line 349): You describe here that your results align with similar recent international studies, but how does it compare with other Dutch studies, for example:

o Hendriks et al. Cost-effectiveness of a multidisciplinary fall prevention program in community-dwelling elderly people: a randomized controlled trial (ISRCTN 64716113). Int J Technol Assess Health Care. 2008 Spring;24(2):193-202. doi: 10.1017/S0266462308080276. PMID: 18400123.

o Peeters et al. Multifactorial evaluation and treatment of persons with a high risk of recurrent falling was not cost-effective. Osteoporos Int. 2011 Jul;22(7):2187-96. doi: 10.1007/s00198-010-1438-4.

As the health care system and access to services has a great influence on costs, comparison of results with studies within the same/similar health care system may be more relevant than comparison with studies in different countries/health care systems.

• Discussion (p. 21, line 378): If I interpret this sentence correctly, then you describe here that the sample was representative of the population that typically receives the In Balance intervention. The level of education in this sample was quite high. Does this reflect the general population that receives In Balance? Might this suggest that extra efforts are needed to reach groups with lower levels of education?

• Discussion (p. 21, lines 367-390): “Moreover, although we used an optimal number of …. of the In Balance intervention.” Apologies but I do not understand this sentence or what is meant here. Which groups are you talking about here? Weren’t there 131 and 133 participants in the intervention and control groups, respectively? Why 12 per group? Can you please rephrase to clarify the point you are making here?

Reviewer #2: Dear Editor,

I write to submit my comments on the manuscript titled “The cost-effectiveness of the Dutch In Balance fall prevention intervention compared to exercise recommendations among community-dwelling older adults with an increased risk of falls: a randomized controlled trials”

The study, via a randomized controlled trial, evaluated the cost-effectiveness of the In Balance fall prevention intervention for community-dwelling older adults with an increased risk of falls compared to general physical activity recommendations (control) from a societal perspective.

Here are my comments

The outcome measures are the number of falls, fall-related injuries, and Quality-Adjusted Life Years (QALYs) based on the EuroQol Five-level questionnaire (EQ-5D-5L) and the Adult Social Care Outcomes Toolkit (ASCOT).

1) Clarification: The authors indicated that the control group received written general recommendations on physical activity via a flyer, in accordance with the World Health Organization's physical activity guidelines and the intervention group received the “In Balance”. Is it the case that the intervention group actually received the physical activity in accordance with the World Health Organization's physical activity guidelines plus the “In Balance,” OR they only received just the Inbalance? That is control=A and Intervention=A+B. Kindly provide clarity on that. This is also important because the intervention group may be involved in physical activities similar to those described in the WHO physical activity guidelines.

2) In line 133, the authors indicated outcome measure and mentioned cost but also said also said in line 159 that the primary outcome measures in this trial were the number of falls and the number of injuries due to a fall

3) Authors indicated that multiple imputation by chained equations were used but authors did not provide any information on the missing mechanism assumption (missing at random, missing completely at random and missing not at random) and justification as to the choice of a selected missing mechanism

4) What was the measurement scale of the outcome measure? Number of falls for instance is a count outcome measure and cost may be continuous. How did the authors analyze the count outcomes? Were they treated as continuous to fit Zellner's seemingly unrelated regression? Or were they treated as counts and fitted Poisson or negative binomial models?. I think more clarity must be provided regarding the measurement scale of the outcome measures, especially when in some cases is cost and in other cases is number of falls etc. The appropriate regression models must be applied in each case.

5) Authors must clarify if they employed a robust standard error estimation in fitting the seemingly unrelated regression. The standard error estimation will affect the confidence intervals of the effect size estimates

6) Authors must present results on complete case analysis as part of the sensitivity analysis to help determine how the results vary compared to pooled effect estimates from the multiple imputation by chained equations

**Do you want your identity to be public for this peer review?** For information about this choice, including consent withdrawal, please see our Privacy Policy

Reviewer #1: No

Reviewer #2: No

---

## [Author Response · Author response to Decision Letter 1]

28 Oct 2025

Dear dr. Chenette,

Thank you for your message. We have revised the Data Availability Statement accordingly. The updated version now reads as follows:

'Only the researchers have access to the data due to privacy considerations, as the dataset contains potentially identifying participant information. An anonymised version of the data is available upon request from prof. dr. Mirjam Pijnappels (m.pijnappels@vu.nl, +31205988467). Prof. dr. Pijnappels served as the Principal Investigator, project coordinator, and project leader of the project during which this study was conducted.'

Thank you for your consideration of our manuscript. We are looking forward to your response.

On behalf of all authors,

Yours sincerely,

Prof. dr. Judith Bosmans

Dear dr. Koleva-Kolarova,

Thank you for giving us the opportunity to revise our manuscript entitled ‘The cost-effectiveness of the Dutch In Balance fall prevention intervention compared to exercise recommendations among community-dwelling older adults with an increased risk of falls: a randomized controlled trial’ and the willingness to consider our manuscript for publication in PLOS ONE. We appreciate the fast review process concerning our manuscript. Please find our responses to the reviewers' comments below. The phrases in italics are changed or added in the revised manuscript. We want to thank the reviewers for their comments which improved our manuscript. We hope that the revised manuscript is acceptable for publication.

Comment 1: Please ensure that your manuscript meets PLOS ONE's style requirements, including those for file naming. The PLOS ONE style templates can be found at

Response and action: We have reviewed the PLOS ONE style requirements and updated the manuscript and file naming accordingly, using the provided templates.

Comment 2: Thank you for stating the following in the Competing Interests section:

“The first authors (JD and MvG) are early-stage investigators employed by the Vrije Universiteit Amsterdam. MvG coordinated the data collection and JD performed the analysis of the data. JD, MvG, PV, DB, BV, MP and JB have no competing interests. One author (BO) is employed by VeiligheidNL, which is the institute that owns the In Balance intervention. VeiligheidNL helped in designing the study, but was not involved in the data analysis and did not have an influence on the results in any way. No funding has been received from a commercial organization.”

We note that one or more of the authors are employed by a commercial company: name of commercial company.

Comment 2a: Please provide an amended Funding Statement declaring this commercial affiliation, as well as a statement regarding the Role of Funders in your study. If the funding organization did not play a role in the study design, data collection and analysis, decision to publish, or preparation of the manuscript and only provided financial support in the form of authors' salaries and/or research materials, please review your statements relating to the author contributions, and ensure you have specifically and accurately indicated the role(s) that these authors had in your study. You can update author roles in the Author Contributions section of the online submission form.

Response and action: We would like to point out that VeiligheidNL is not a commercial company, but a non-profit knowledge institute with the status of a Public Benefit Organisation. They are member of Eurosafe, the European Association of Injury Prevention and Safety Promotion and aim to reduce the number of accidents and injuries in the Netherlands. As such, fall prevention is one of their key themes for injury prevention and public safety (https://www.veiligheid.nl/about-us).

To clarify, we adapted the Funding Statement to: ‘This research was funded by a research grant (#555002018) from the Netherlands Organisation for Health Research and Development (ZonMw). The funder provided support in the form of salaries for authors MvG, PV, DB, BV, MP and JB and study costs, but did not have any additional role in the study design, data collection and analysis, decision to publish, or preparation of the manuscript. The specific roles of these authors are articulated in the ‘author contributions’ section. One author (BO) is employed by the VeiligheidNL foundation, which is a non-profit public benefit organisation and a knowledge institute for injury prevention and public safety that owns the In Balance intervention. VeiligheidNL participated in designing the study, but did not have any additional role in the data collection and analysis, decision to publish, or preparation of the manuscript. No funding has been received from any commercial organization.’

Comment 2b: Please also provide an updated Competing Interests Statement declaring this commercial affiliation along with any other relevant declarations relating to employment, consultancy, patents, products in development, or marketed products, etc.

Response and action: We adapted the Competing Interests Statement to: ‘The first authors (JD and MvG) are early-stage investigators employed by the Vrije Universiteit Amsterdam. MvG coordinated the data collection and JD performed the analysis of the data. JD, MvG, PV, DB, BV, MP and JB have no competing interests. One author (BO) is employed by the VeiligheidNL foundation, which is a non-profit public benefit organisation and a knowledge institute for injury prevention and public safety that owns the In Balance intervention. VeiligheidNL participated in designing the study, but did not have any additional role in the data collection and analysis, decision to publish, or preparation of the manuscript. This does not alter our adherence to PLOS ONE policies on sharing data and materials.’. The updated Funding Statement and Competing Interests Statement are included in our cover letter.

Comment 3: In the online submission form, you indicated that [Insert text from online submission form here].

Response and action: This study involves human participant data that cannot be publicly shared due to ethical and privacy considerations in accordance with the protocol approved by the research ethics board. Therefore, the full dataset is not publicly available but can be shared upon reasonable request. Data requests will be assessed on a case-by-case basis to ensure that participant confidentiality is safeguarded. Requests can be directed to m.pijnappels@vu.nl. Relevant data supporting the findings are included in the manuscript.

Comment 4: Thank you for stating the following in the Acknowledgments Section of your manuscript:

“We want to thank all participants and therapists for participating in this study. Moreover, this research was funded by a research grant (#555002018) from the Netherlands Organisation for Health Research and Development (ZonMw). The funders had no role in study design, data collection and analysis, decision to publish, or preparation of the manuscript. We also want to thank VeiligheidNL, in particular Sanne Frazer, for their help with designing this study. “

“This research was funded by a research grant (#555002018) from the Netherlands Organisation for Health Research and Development (ZonMw). The funders had no role in study design, data collection and analysis, decision to publish, or preparation of the manuscript.”

Response and action: We removed the funding information from the Acknowledgements Section, which now reads: ‘We want to thank all participants and therapists for participating in this study. We also want to thank VeiligheidNL, in particular Sanne Frazer, for their help with designing this study.’. For the adapted Funding Statement, please refer to our response to comment 2a.

Comment 5: Your ethics statement should only appear in the Methods section of your manuscript. If your ethics statement is written in any section besides the Methods, please delete it from any other section.

Response and action: Thank you for the clarification. We have revised the manuscript to ensure that the ethics statement now appears exclusively in the Methods section, and have removed it from all other sections.

Comment 6: Please include captions for your Supporting Information files at the end of your manuscript, and update any in-text citations to match accordingly. Please see our Supporting Information guidelines for more information: http://journals.plos.org/plosone/s/supporting-information.

Response and action: We have now added captions for all Supporting Information files at the end of the manuscript, in accordance with PLOS ONE’s guidelines. Additionally, we have reviewed and updated all in-text citations to ensure they correctly correspond to the revised Supporting Information section.

Additional editor comments:

General comment: Dear Authors, Thank you for submitting your manuscript to Plos One. I would like to state that this is a well-conducted economic analysis and a well-written article which I enjoyed reading. I have now reached a decision which is Major Revision. In addition to the comments from the reviewers, please see below my comments:

Response: Thank you for your positive feedback and for the opportunity to revise our manuscript. We have carefully addressed your comments as well as those from the reviewers.

Comment 1: Could the authors please justify not including non-Dutch speakers in the study?

Response and action: Non-Dutch speakers were not included in the study because In Balance is a Dutch program, delivered entirely in Dutch and accompanied by Dutch-language materials.

We included this in the Methods section of the manuscript on page 7 as follows: ‘Other exclusion criteria were the inability to walk 100 meters independently or to perform activities of daily living without assistance, cognitive impairment defined as a score lower of 18 or lower on the Mini-Mental State Examination [1], self-reported contraindications (i.e., cardiovascular, neurological and orthopedic problems), participation in a fall prevention intervention within the last 6 months, or insufficient proficiency in Dutch, as the In Balance intervention and all materials were only available in Dutch.’

Comment 2: Similarly to the comment from another reviewer, could the authors please clarify why they considered productivity losses in a population group of elderly and potentially retired participants? Could the authors please elaborate if excluding productivity losses would have an impact on cost-effectiveness results?

Response and action: Although the majority of participants were retired since the official retirement age in the Netherlands is 67 years, our inclusion criterion was being aged 65 years and older, and 14 participants in our study were still employed. Additionally, 22 participants reported being unable to perform unpaid work due to health complaints. To ensure a broad, comprehensive societal assessment of costs, we chose to include productivity losses, both from paid and unpaid work, in our analysis. From a societal perspective, these costs are relevant and should be taken into account. Moreover, our data confirm that some participants were still working at the time of the study. Exclusion of these costs results in slightly lower total cost estimates, but does not change the overall conclusions regarding the cost-effectiveness of the In Balance program as shown by our sensitivity analysis from the healthcare perspective.

We included this in the Methods section of the manuscript on page 8 as follows: ‘Although most participants were retired, 14 were still employed and 22 reported being unable to perform unpaid work due to health issues. Given the Dutch recommendation to take a broad societal perspective, productivity losses, both paid and unpaid, were included to provide a comprehensive estimate of costs [2].’

General comment: Thank you. Good luck with revising the article.

Response: Thank you. We did our best to address all comments thoroughly in the revision.

Reviewer 1:

General comment: This manuscript describes a well-conducted economic evaluation of the In Balance intervention compared with physical activity recommendations in non-frail and pre-frail older adults with a high fall risk. The manuscript reads well and the provided information is clear. Findings support the conclusion. Some clarifications would be helpful on methodological choices that were made.

Response: We thank the reviewer for the positive and encouraging feedback. We appreciate your recognition that the manuscript reads well, the information provided is clear, and that the findings support the conclusion. In response to your suggestion, we have added clarifications on specific methodological choices. We hope these additions improve the transparency of our methods.

Comment 1: Introduction (p. 4, line 65): You say that In Balance is a widely implemented intervention. Is it widely implemented in the Netherlands or Europe or the world? In line 74 you say that it is important to evaluate the cost-effectiveness to inform policy makers to optimally allocate resources or to arrange reimbursement. Given that it is already implemented, how is the intervention financed now? Are there concerns regarding implementation (e.g. are the right people reached)? In other words, why is it important to do the cost-effectiveness evaluation now that the intervention is already implemented?

Response and action: Thank

---

## [Decision Letter · Decision Letter 1]

9 Dec 2025

The cost-effectiveness of the Dutch In Balance fall prevention intervention compared to exercise recommendations among community-dwelling older adults with an increased risk of falls: a randomized controlled trial

PONE-D-25-05745R1

Dear Prof Bosmans,

We’re pleased to inform you that your manuscript has been judged scientifically suitable for publication and will be formally accepted for publication once it meets all outstanding technical requirements.

I would, however, suggest that you amend the phrase 'usual care condition' used in the Introduction to 'usual care' as I believe you are referring to the control group that received 'usual care' rather than the physical condition itself.

Kind regards,

Rositsa Koleva-Kolarova

Academic Editor

PLOS One

Additional Editor Comments (optional):

I would, however, suggest that you amend the phrase 'usual care condition' used in the Introduction to 'usual care' as I believe you are referring to the control group that received 'usual care' rather than the physical condition itself.

Reviewers' comments:

Reviewer's Responses to Questions

**Comments to the Author**

Reviewer #2: All comments have been addressed

2. Is the manuscript technically sound, and do the data support the conclusions?

Reviewer #2: Yes

3. Has the statistical analysis been performed appropriately and rigorously?

Reviewer #2: Yes

4. Have the authors made all data underlying the findings in their manuscript fully available?

Reviewer #2: Yes

5. Is the manuscript presented in an intelligible fashion and written in standard English?

Reviewer #2: Yes

Reviewer #2: The authors have adequately responded to all my previous comments and have provided clarity in the manuscripts

**Do you want your identity to be public for this peer review?** For information about this choice, including consent withdrawal, please see our Privacy Policy

Reviewer #2: No

---

## [Editor Report · Acceptance letter]

PONE-D-25-05745R1

PLOS One

Dear Dr. Bosmans,

I'm pleased to inform you that your manuscript has been deemed suitable for publication in PLOS One. Congratulations! Your manuscript is now being handed over to our production team.

Kind regards,

on behalf of

Dr. Rositsa Koleva-Kolarova

Academic Editor

PLOS One